# Attractiveness and retention factors for employed physiotherapists in France: A multicentre survey

**Aurélien Auger**[1,2], **Arnaud Delafontaine**[3], **Jeanne Lecordier**[1], **Thomas Rulleau**[4], **François-Régis Sarhan**[1,5,6]*

**1** Centre Hospitalier de la Risle, Pont-Audemer, France, **2** Institut d'ingénierie de la Santé, UFR de Médecine – Université Picardie Jules Verne, Amiens, France, **3** Université Libre de Bruxelles, Bruxelles, Belgique, **4** Nantes Université, CHU Nantes, Motricité - Interactions - Performance, MIP, Nantes, France, **5** UR 7516 CHIMERE, Université Picardie Jules Verne, Amiens, France, **6** Physiotherapy School, Centre Hospitalier Universitaire Amiens – Picardie, Amiens, France

* francois-regis.sarhan@u-picardie.fr

## Abstract

### Objectives

The aim of this study was to investigate the factors influencing the attractiveness of employed physiotherapy, considering the shortage of physiotherapists in the salaried sector, previously unsuccessful measures.

### Methods

A survey was conducted using an online self-administered questionnaire. The primary outcome measure involved assessing respondents' agreement with each statement in key domains. Regarding the secondary outcome measures, the relationships between respondents' age and their responses, as well as the relationship between their profession and their responses, were examined.

### Results

The study included 448 participants in France, encompassing both primary and secondary care settings. The sample consisted of practicing physiotherapists (83.1%) and healthcare managers (16.9%). Respondents expressed significant concerns regarding the attractiveness of employed physiotherapy. For the primary outcome, the study identified potential solutions, including improving financial recognition, promoting continuous professional development, and enhancing inclusion in healthcare projects. For the secondary outcomes, statistically significant relationships were found between respondents' age and their answers for variables 16 (additional qualifications; $p = 0.048$), 24 (work-life balance; $p = 0.010$), and 25 (professional quality of life; $p = 0.0005$).

### Conclusion

The findings underscore the need to enhance financial recognition, support continuous professional development, and increase inclusion in healthcare projects to effectively

**Data availability statement:** All relevant data are within the manuscript and its Supporting Information files.

**Funding:** The author(s) received no specific funding for this work.

**Competing interests:** The authors have declared that no competing interests exist.

improve the attractiveness of employed physiotherapy. These results have significant implications for policymakers and healthcare institutions grappling with recruitment challenges. Further research and collaborative efforts are crucial to implementing the recommended measures and assessing their impact on the employed physiotherapy profession.

## Introduction

In 2020, there were 91,485 practicing physiotherapists in France, with approximately 14.7% to 17.0% of them employed in the public sector, working in hospitals, rehabilitation centers, and other healthcare institutions [1,2]. These physiotherapists play an essential role in providing care to hospitalized patients and those undergoing rehabilitation [3,4].

However, the healthcare sector in France is currently facing significant challenges in meeting the growing demand for physiotherapy services [5]. A 2021 report from the Technical Agency for Information on Hospitalization [6] identified the profession of physiotherapist as "sensitive" in terms of recruitment, noting that over 40% of the responding establishments in their survey reported difficulties in hiring. In addition to the major role played by physiotherapists in recovery and maintaining patient mobility, the system is under increasing pressure. This is driven by an aging population requiring more complex care, rising healthcare costs, and an insufficient number of qualified physiotherapists in public sector positions [7–9]. These factors exacerbate the strain on public healthcare institutions and underscore the urgent need to address workforce challenges.

The attractiveness and retention of physiotherapists in the public sector have become critical issues in France [2,5]. Several factors contribute to these challenges. One major issue is the wage disparity between public and private practice physiotherapists. In 2022, physiotherapists in private practice earned an average of €44,243, 30–50% [10] more than their counterparts employed in the public sector, who earned €32,808 on average [11]. This wage gap creates a strong incentive for physiotherapists to choose private practice over public sector employment [12]. In addition to financial considerations, public sector physiotherapists often face professional dynamics, management practices, and limited career advancement opportunities, all of which can further deter potential recruits and contribute to high turnover rates [5,13–15].

Moreover, the aging workforce, combined with the increasing number of retirements among experienced physiotherapists, exacerbates the shortage in the public sector [16,17]. These factors, alongside the demanding nature of the work and the high patient load in public hospitals, make attractiveness even more challenging.

Between 2016 and 2021, ministerial measures were introduced to enhance salary scales and implement commitment bonuses for public sector employees. However, the effectiveness of these initiatives in addressing attractiveness and retention issues has not been formally assessed [18,19]. Despite these efforts, attractiveness and retention challenges persist. The limited success of these measures suggests that a more comprehensive and sustained approach is needed [20,21]. Policies have often focused more on nursing staff, with less emphasis placed on physiotherapists, which may have contributed to the ongoing difficulties in the physiotherapy workforce [19]. At the institutional level, there is an opportunity need to develop localized strategies to improve fidelity and attractiveness and to tackle these issues effectively. Given the recruitment challenges in the profession, institutions might consider implementing approaches that emphasize project-based solutions and create a supportive environment for students. Since 2015, French physiotherapists have been trained over five years, earning a Master's degree. This reform has led to a substantial increase in their competencies and responsibilities. However, these advancements in qualifications and professional scope have

not been matched by corresponding changes in their roles, responsibilities, or compensation within the public sector [22–24].

The aim of this study was to explore the factors that influence the attractiveness of salaried physiotherapy positions in the public sector, particularly those related to attractiveness and retention. Specifically, the study aims to identify the key barriers that prevent physiotherapists from choosing public sector employment and to understand the factors that contribute to job dissatisfaction and high turnover rates. This study will also examine the role of compensation, career development opportunities, work-life balance, and institutional management practices in shaping physiotherapists' decisions to remain in or leave the public sector.

Based on the findings, the study will propose specific recommendations for improving the attractiveness and retention of physiotherapists in public healthcare institutions. It is hypothesized that addressing the wage gap, improving career development opportunities, and fostering better institutional support will significantly enhance the attractiveness of public sector physiotherapy positions and reduce turnover rates.

## Materials and methods

This study employed a cross-sectional quantitative survey design using Likert-type scale responses. The survey was designed, in line with STROBE guideline [25], by a team consisting of a physiotherapy manager, a physiotherapy lecturer, and a senior physiotherapist.

The study took place within the context of French public hospitals where physiotherapists are employed. Data collection occurred between May 22 and June 21, 2021, and the analysis took place from June 22 to July 16, 2021.

The study targeted physiotherapists currently employed in public hospitals in France. Exclusion criteria included physiotherapists working in private practice, physiotherapy students, and those working in private clinics or non-public institutions. Additionally, responses from other healthcare professionals, such as managers from nursing or other rehabilitation professions, were excluded to ensure the focus remained on physiotherapists. Incomplete or inconsistent questionnaires and participants who withdrew consent were also excluded.

In 2020, the estimated number of physiotherapists employed in the public sector in France was 15,550 [1,2]. The sample size was determined based on statistical considerations to ensure the reliability of the results [26]. A target of 375 responses was set to achieve a 5.0% margin of error and a 95% confidence interval. Additionally, a comparison with known demographic data was conducted to verify the representativeness of the sample, ensuring that the findings could be generalized to the broader population of physiotherapists in the public sector.

The primary outcome of the study focused on respondents' level of agreement with statements in four key domains that reflect various factors influencing the attractiveness and retention of physiotherapists in public hospitals. These domains are assessed using Likert scale data, with responses ranging from "strongly agree" to "strongly disagree." Secondary outcomes involved examining the relationships between demographic variables, such as age and profession, and the responses to the survey items.

Data were collected using an anonymous, self-administered questionnaire hosted on an online platform (*Framaforms* - Framasoft; Lyon). The questionnaire (Table 1) was designed to gather demographic information and explore physiotherapists' perceptions and experiences regarding their institutions. The survey was designed based on an extensive review of existing literature on workforce attractiveness and retention in healthcare, specifically within the physiotherapy field [27–29]. Additionally, consultations were held with a team of experts, including physiotherapy managers, lecturers, and senior physiotherapists. This ensured that the survey reflected current issues and challenges faced by physiotherapists in public hospitals. Thus, it included statements related to factors influencing the attractiveness of physiotherapy

**Table 1. Questionnaire of the survey.**

*PART I: General demographic questions*

*PART II: Feelings and Experiences in Daily Work*
*Question: "Do you agree with the following statement?"*
*Response options: "Strongly agree," "Agree," "Neutral," "Disagree," "Strongly disagree".*
1. My institution faces an issue with the attractiveness of physiotherapist positions.
2. The attractiveness measures implemented by successive governments since 2016 have been sufficient.
3. You experience difficulties or feel overwhelmed in your daily work.

*PART III: Opinion About Attractiveness Measures*
*Question: "Do you agree with the following statement?"*
*Response options: "Strongly agree," "Agree," "Neutral," "Disagree," "Strongly disagree."*
**Domain 1: Adoption of National Measures**
1. Wage increases enhance employee attractiveness.
2. Recognizing a Master's degree for physiotherapy studies enhances employee attractiveness.
3. Recognizing additional diplomas in the public hospital service enhances employee attractiveness.
**Domain 2: Inclusion in Daily Structure**
4. Including physiotherapists in the development of rehabilitation projects enhances employee attractiveness.
5. Including physiotherapists in shared inter-facility projects enhances employee attractiveness.
6. Including physiotherapists in the management of their facility enhances employee attractiveness.
**Domain 3: Supervision of Employed Physiotherapists**
7. Having a healthcare manager from one of the rehabilitation professions enhances employee attractiveness.
8. Having a senior healthcare executive from one of the rehabilitation professions enhances employee attractiveness.
9. Entrusting cross-functional activities to physiotherapists (e.g., ethics, steering commissions, working groups) enhances employee attractiveness.
**Domain 4: Quality of Life and Mentorship**
10. Involving physiotherapists in tutoring trainees enhances employee attractiveness.
11. The quality of work life is an important factor in choosing your professional activity.
12. Management consideration of personal quality of life promotes employee attractiveness.

employment in public hospitals. These factors covered a range of topics, from the adoption of national measures to the inclusion of physiotherapists in daily operations, their supervision, and the quality of life and mentorship available to them. These four domains were chosen to provide a holistic view of the factors influencing the attractiveness retention of physiotherapists in the public sector. By exploring both national-level policies and the daily realities of physiotherapists in hospitals, we aimed to identify opportunities for improvement at every level. The survey thus sought to determine whether local practices could independently influence retention and attractiveness, independent of broader government measures.

To minimize bias, participation was voluntary and the survey was designed to be anonymous. Efforts were made to ensure that the questionnaire was clear and relevant, with a pilot test conducted to refine the survey items.

Quantitative data, including respondents' age and professional experience, were analyzed using descriptive statistics. The Likert scale responses were grouped and analyzed to determine the frequency of agreement with each statement, providing insights into the factors affecting the attractiveness of physiotherapy positions in public hospitals.

Descriptive statistics were applied to analyze both demographic and Likert scale data. For secondary outcomes, a chi-square test was conducted to assess the relationship between demographic factors, such as age and profession, and responses to the survey questions. Statistical analysis was performed using XLSTAT 2021.2.2 software (Addinsoft 2021), with a significance threshold set at $p < 0.05$. Missing data were addressed by including only complete responses in the analysis, without imputation methods.

The study adhered to ethical guidelines and was registered with the French data protection authority (CNIL) under registration number 2223050. Informed consent was obtained from all participants before they began the survey. Participants were also informed that they could

withdraw their consent at any time up to the end of the data collection process. Validation renders the data anonymous, making it impossible to erase it afterwards.

## Results

A total of 460 individuals participated in the survey. Of these, 453 responses were deemed eligible for analysis, and after data cleaning and processing, a final dataset of 448 responses was analyzed (Fig 1). Non-participation occurred due to incomplete responses, technical issues during survey submission, or the exclusion of respondents who were not physiotherapists, such as managers from nursing or other rehabilitation professions.

The final sample included 76.8% women and 23.2% men, reflecting a slight overrepresentation of women compared to the national sex ratio of 69.4% women and 30.6% men in the national physiotherapy workforce data. The mean age of participants was 40.1 years (SD 11.3 years). Among respondents, 83.1% were practicing physiotherapists, while 16.9% held managerial or leadership positions. There was no significant age differences between professional subcategories ($p > 0.05$; Table 2).

Participants were geographically distributed across all French regions. Regarding healthcare settings, 30.4% worked in university hospitals, 26.6% in general hospitals and 12.3% in other public healthcare centers, such as healthcare and social services institutions or specialized facilities. Most respondents (71.2%) worked full-time, while 24.3% held part-time or non-full-time positions, and 4.5% combined employed and private practice work. The average professional experience was 9.7 years. Additionally, 54.2% of participants reported having no additional qualifications, while the remainder held at least one *(such as Master, PhD or Specialization)*.

Regarding workplace challenges, 49.3% of respondents reported feeling "often" or "always" overwhelmed in their daily work (Table 3). Most participants (79.9%) agreed or strongly agreed that their institution faced issues of attractiveness. In terms of government measures, 64.1% of respondents disagreed or strongly disagreed that initiatives since 2016 were sufficient (Table 4).

Analysis of *the four domains* revealed key factors affecting physiotherapy emplo*yment attractiveness (*Table 4*).

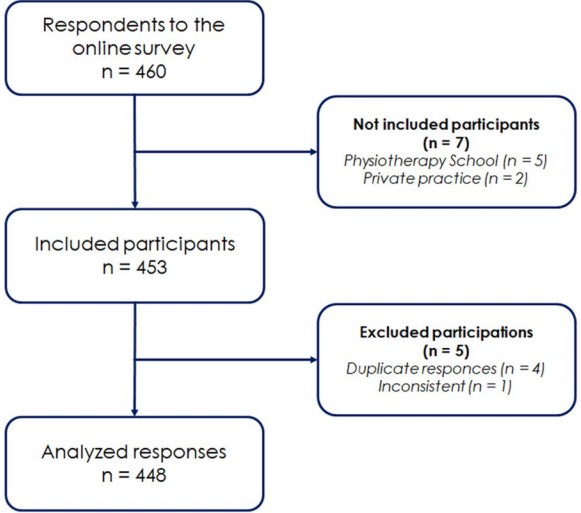

**Fig 1. Study flow chart.**

**Table 2. Respondents' age (n = 448). No significant difference between the subgroups was identified (p > 0.05).**

|  | n (%) | Age (years +/− SD) | Sex-ratio (% women/men) |
|---|---|---|---|
| **National data** (1,2) **(only practicing physiotherapists)** | 15 550 (1) | 43.0 (2) | 69.4%/ 30.6% (1) |
| **All respondents** | 448 | 40.1 (+/− 11.33) | 76.8%/ 23.2% |
| **Practicing physiotherapists** | 371 (82.8%) | 38.5 (+/− 10.9) | 78.7%/ 21.3% |
| **Healthcare managers** | 58 (12.9%) | 46.8 (+/− 9.5) | 67.2%/32.8% |
| **Senior healthcare executives** | 14 (3.1%) | 53.2 (+/− 9.7) | 71.4%/28.6% |
| **Director of Patient Care Services** | 5 (1.1%) | 49.2 (+/− 11.6) | 60.0%/ 40.0% |

**Table 3. Difficulties or overwhelmed in daily work (n = 448).**

|  | Always | Often | Sometimes | Rarely | Never |
|---|---|---|---|---|---|
| You experience difficulties or feel overwhelmed in your daily work. | 10.3 [8.7–11.9] | 39.1 [37.5–40.7] | 38.8 [37.2–40.4] | 11.2 [9.6–12.8] | 0.7 [−0.9–2.3] |

Data are expressed in percent and 95% confidence intervals [CI].

**Table 4. Agreement of respondents with statements (n = 448).**

|  | Strongly agree | Agree | Neither disagree nor agree | Disagree | Strongly disagree |
|---|---|---|---|---|---|
| The attractiveness measures implemented by successive governments since 2016 have been sufficient. | 4.5 [3.1–5.9] | 6.7 [5.3–8.1] | 24.8 [23.3–26.2] | 42.6 [41.2–44.1] | 21.4 [20.0–22.9] |
| Wage increases enhance employee attractiveness. | 48.9 [46.7–51.0] | 41.1 [38.9–43.2] | 6.5 [4.3–8.6] | 2.7 [0.5–4.8] | 0.9 [−1.2–3.0] |
| Recognizing a Master's degree for physiotherapy studies enhances employee attractiveness. | 21.9 [20.9–22.8] | 26.3 [25.4–27.3] | 32.4 [31.4–33.3] | 13.8 [12.9–14.8] | 5.6 [4.6–6.6] |
| Recognizing additional diplomas in the public hospital service enhances employee attractiveness. | 25.2 [24.0–26.5] | 37.1 [35.8–38.3] | 25.2 [24.0–26.5] | 6.5 [5.2–7.7] | 6.0 [4.8–7.3] |
| Including physiotherapists in the development of rehabilitation projects enhances employee attractiveness. | 50.2 [48.0–52.4] | 40.4 [38.2–42.6] | 7.8 [5.6–10.0] | 0.7 [−1.5–2.8] | 0.9 [−1.3–3.1] |
| Including physiotherapists in shared inter-facility projects enhances employee attractiveness. | 31.7 [30.1–33.3] | 36.6 [35.0–38.2] | 28.1 [26.6–29.7] | 2.7 [1.1–4.2] | 0.9 [−0.7–2.5] |
| Including physiotherapists in the management of their facility enhances employee attractiveness. | 24.8 [23.1–26.4] | 43.3 [41.6–45.0] | 27.9 [26.2–29.6] | 3.6 [1.9–5.2] | 0.4 [−1.2–2.1] |
| Having a healthcare manager from one of the rehabilitation professions enhances employee attractiveness. | 34.8 [33.3–36.4] | 38.8 [37.3–40.4] | 18.8 [17.2–20.3] | 6.5 [4.9–8.0] | 1.1 [−0.4–2.7] |
| Having a senior healthcare executive from one of the rehabilitation professions enhances employee attractiveness. | 26.8 [25.4–28.1] | 31.7 [30.3–33.1] | 33.0 [31.7–34.4] | 7.1 [5.8–8.5] | 1.3 [−0.0–2.7] |
| Entrusting cross-functional activities to physiotherapists | 28.8 [27.1–30.5] | 44.9 [43.2–46.5] | 21.0 [19.3–22.7] | 4.9 [3.2–6.6] | 0.4 [−1.2–2.1] |
| Involving physiotherapists in tutoring trainees enhances employee attractiveness. | 23.2 [21.8–24.6] | 38.2 [36.8–39.6] | 29.5 [28.1–30.9] | 7.6 [6.2–9.0] | 1.6 [0.2–3.0] |
| The quality of work life is an important factor in choosing your professional activity. | 80.8 [77.6–84.0] | 18.1 [14.9–21.3] | 0.7 [−2.6–3.9] | 0.4 [−2.8–3.7] | 0.00 [−3.2–3.2] |
| Management consideration of personal quality of life promotes employee attractiveness. | 63.8 [61.3–66.4] | 30.8 [28.2–33.4] | 4.7 [2.1–7.2] | 0.4 [−2.1–3.0] | 0.2 [−2.3–2.8] |

Data are expressed in percent and 95% confidence intervals [CI].

### National measures

Respondents indicated that increasing salaries (89.9%) and recognizing additional qualifications (62.3%) would significantly enhance attractiveness. Opinions on the value of a master's degree for physiotherapy studies were more mixed, with 48.2% agreeing or strongly agreeing.

### Healthcare institutions

The inclusion of physiotherapists in rehabilitation projects (90.6%) and shared inter-facility initiatives (68.3%) were perceived as improving attractiveness. Proper management involvement was also considered beneficial (68.1%).

### Management

The presence of a clinical supervisor or senior healthcare executive with initial training in rehabilitation was viewed positively, with 73.6% and 58.5% agreement, respectively. Involvement in cross-functional activities, such as ethics committees or steering commissions, was also seen as advantageous (73.7%).

### Individual factors

Work-life balance and professional quality of life emerged as critical factors, with 80.8% and 94.6% agreement, respectively. Mentorship was less emphasized, with only 61.4% perceiving it as a key factor.

Statistical analysis revealed significant relationships between age and certain variables, particularly for participants under 40 years old, who placed greater importance on the recognition of additional qualifications, work-life balance in career choices, and quality of life considerations in assessing job attractiveness (Table 5). Conversely, no significant associations were observed between others measured variables

## Discussion

### Key results

The aim of this study was to explore the factors influencing the attractiveness and retention of physiotherapists employed in French public hospitals. The findings underscored the multi-faceted nature of this challenge, revealing significant barriers and opportunities for improvement. Key results highlighted the critical role of quality of work-life, professional recognition, and institutional policies in shaping the decisions of physiotherapists to remain in the public sector. Importantly, respondents emphasized the value of work-life balance, teamwork, and the recognition of additional qualifications. Advantages such as job security, flexible working

**Table 5. Effect of age on socio-professional outcomes (n = 448). For statistical analysis, age was dichotomized into two categories: < 40 years and ≥ 40 years. The chi-square test was used to assess differences between these groups regarding the stated variables.**

| Age group n (%) | Strongly agree | Agree | Neither disagree nor agree | Disagree | Strongly disagree | Chi-Squared Tests |
|---|---|---|---|---|---|---|
| **The quality of worklife is important in the choice of your professional activity** | | | | | | |
| <40 | 221 (89.5%) | 25 (10.1%) | 1 (0.4%) | 0 (0.0%) | 0 (0.0%) | X² = 27.4 |
| ≥40 | 141 (70.1%) | 56 (27.9%) | 2 (1.0%) | 2 (1.0%) | 0 (0.0%) | p < .001 |
| **Taking personal quality of life into account by management promotes attractiveness** | | | | | | |
| <40 | 177 (71.7%) | 62 (25.1%) | 8 (3.2%) | 0 (0.0%) | 0 (0.0%) | X² = 17.2 |
| ≥40 | 109 (54.2%) | 76 (37.8%) | 13 (6.5%) | 2 (1.0%) | 1 (0.5%) | p = $1.7 \times 10^{-3}$ |

hours, and opportunities for professional development were acknowledged, but these were often overshadowed by systemic shortcomings, including inadequate compensation and limited involvement in institutional decision-making processes.

### Interpretation

The results provide a nuanced understanding of the factors influencing the attractiveness of physiotherapy positions in public hospitals. Consistent with prior studies [30–32], this research highlights the pivotal role of professional recognition and work-life balance in enhancing job satisfaction and retention. However, the study also reveals a disconnect between the advanced qualifications of physiotherapists and the institutional recognition of these competencies. Despite the implementation of a Master's degree for physiotherapists in France since 2015 [15], public institutions have yet to fully integrate this elevated status into their operational and managerial frameworks. This misalignment not only undermines professional morale but also hampers the efficiency of healthcare delivery [33,34]. The findings also emphasize the importance of involving physiotherapists in cross-functional roles and institutional projects, suggesting that such inclusion could significantly improve job satisfaction and professional commitment [3,4,34]. Moreover, the generational shift in workforce priorities, particularly among younger physiotherapists, underscores the need for policies that address work-life balance and career development opportunities [18,30].

### Barriers to attractiveness and retention

The study revealed several barriers that diminish the appeal of physiotherapy positions in public hospitals. Inadequate compensation and lack of recognition for advanced qualifications emerged as critical issues. Respondents under 40 years old were particularly sensitive to the lack of career development opportunities and the limited integration of physiotherapists into institutional decision-making. These findings align with prior literature emphasizing the importance of aligning professional roles with advanced training and qualifications [35–37].

### Institutional and policy measures

Institutional policies and management practices significantly influence physiotherapists' job satisfaction. Respondents highlighted the need for policies that prioritize work-life balance, mentorship, and career growth. The study also found that institutions often fail to fully use the skills of physiotherapists, particularly those with advanced qualifications, leading to frustration and reduced retention rates. While financial revaluations and attractiveness bonuses have had limited success when implemented in isolation, they remain essential as part of a broader, systemic reform. A structured salary revision, combined with institutional policies recognizing additional qualifications and improving working conditions, could help address the deeper structural challenges of the profession. This underscores the need for more targeted and systemic interventions that address the deeper, structural challenges specific to the profession [38–40].

### Recommendations

Based on the survey findings and existing literature [41,42], the study proposed ten actionable measures to improve the appeal of physiotherapy positions in public hospitals (Table 6). These measures include a reassessment of salary policies to enhance long-term career attractiveness and reduce turnover, recognition of additional qualifications, and the development of institutional policies supporting rehabilitation professions.

**Table 6. 10 measures to improve the appeal of physiotherapy positions in public hospitals.**

| Measure No. | Measure Title |
| --- | --- |
| 1 | Revise salaries for employed physiotherapists in accordance with their level of education. |
| 2 | Recognize additional diplomas and establish a hospital-university status. |
| 3 | Develop an institutional policy supporting rehabilitation professions, merging rehabilitation projects with tailored training plans. |
| 4 | Organize meetings among rehabilitation professionals within the Grouping of Local Hospitals (GLH). |
| 5 | Ensure rehabilitation professionals are supervised by a peer with a background in rehabilitation. |
| 6 | Involve physiotherapists in cross-functional missions within intervention services or at the institutional level. |
| 7 | Allocate dedicated time for supervising interns. |
| 8 | Provide training and information to physiotherapists regarding mentoring interns. |
| 9 | Implement a workplace quality of life policy within the institution and engage physiotherapists in the process. |
| 10 | Facilitate a balance between employees' professional and personal lives. |

Additionally, fostering interdisciplinary collaboration and involving physiotherapists in cross-functional missions were identified as key strategies to improve job satisfaction and retention.

## Limitations

This study has several limitations that should be considered when interpreting the results. First, the survey instrument used was not formally validated, which may introduce biases in the data collection process. Additionally, the online distribution method posed challenges in achieving a representative sample, as participation was voluntary. This self-selection bias likely attracted individuals who already had a vested interest in the topic or specific experiences to share, potentially skewing the findings. Finally, the study's scope was confined to the French public health system, which inherently restricts the applicability of the findings to other healthcare contexts.

This study highlights the pressing need for systemic changes to improve the attractiveness and retention of physiotherapists in French public hospitals. Key recommendations include salary adjustments, professional recognition, and policies supporting work-life balance and mentorship. These measures address barriers such as inadequate compensation and limited career opportunities, aiming to align professional roles with evolving competencies and workforce expectations.

The challenges identified are not unique to France; literature shows that similar issues are prevalent in other countries. The evolving role of physiotherapists and their enhanced qualifications make this an increasingly relevant global concern. Addressing these challenges is essential to ensuring a sustainable workforce and high-quality care in healthcare systems worldwide.

- **Consent to participate:** All participants have given their consent to participate in this study.

- **Consent for publication:** All authors have given their consent for publication.

## Author contributions

**Conceptualization:** Aurélien Auger, François-Régis Sarhan.

**Data curation:** Aurélien Auger, Jeanne Lecordier.

**Formal analysis:** Aurélien Auger, Arnaud Delafontaine, François-Régis Sarhan.

**Funding acquisition:** Jeanne Lecordier, François-Régis Sarhan.

**Investigation:** Aurélien Auger.

**Methodology:** Aurélien Auger, Arnaud Delafontaine, François-Régis Sarhan.

**Project administration:** Jeanne Lecordier, François-Régis Sarhan.

**Resources:** Aurélien Auger, Arnaud Delafontaine, Thomas Rulleau, François-Régis Sarhan.

**Software:** Aurélien Auger.

**Supervision:** Arnaud Delafontaine, Jeanne Lecordier, François-Régis Sarhan.

**Validation:** Aurélien Auger, Arnaud Delafontaine, Jeanne Lecordier, Thomas Rulleau, François-Régis Sarhan.

**Visualization:** Aurélien Auger, Arnaud Delafontaine, François-Régis Sarhan.

**Writing – original draft:** Aurélien Auger, François-Régis Sarhan.

**Writing – review & editing:** Aurélien Auger, Arnaud Delafontaine, Thomas Rulleau, François-Régis Sarhan.

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
