## [Decision Letter · Decision Letter 0]

16 Sep 2024

PONE-D-24-30478Attractiveness and Retention Factors for Employed Physiotherapists in France: A Multicentre SurveyPLOS ONE

Dear Dr. Sarhan,

Thank you for submitting your manuscript to PLOS ONE. After careful consideration, we feel that it has merit but does not fully meet PLOS ONE’s publication criteria as it currently stands. Therefore, we invite you to submit a revised version of the manuscript that addresses the points raised during the review process.

We look forward to receiving your revised manuscript.

Kind regards,

Saravana Kumar

Academic Editor

PLOS ONE

Journal Requirements:

2.Please ensure that you include a title page within your main document. You should list all authors and all affiliations as per our author instructions and clearly indicate the corresponding author.

3. Abstract missing from manuscript

Please amend your manuscript to include your abstract after the title page.

Reviewers' comments:

Reviewer's Responses to Questions

**Comments to the Author**

1. Is the manuscript technically sound, and do the data support the conclusions?

Reviewer #1: Partly

Reviewer #2: Partly

2. Has the statistical analysis been performed appropriately and rigorously? 

Reviewer #1: Yes

Reviewer #2: No

3. Have the authors made all data underlying the findings in their manuscript fully available?

Reviewer #1: Yes

Reviewer #2: Yes

4. Is the manuscript presented in an intelligible fashion and written in standard English?

Reviewer #1: No

Reviewer #2: Yes

5. Review Comments to the Author

Reviewer #1: Thanks you for the opportunity to review this manuscript. The subject is topical and this study has the potential to be of interest to those interest in health workforce development. Whilst the methods were appropriate, and much of the conduct and reporting of the study are commendable, there is weakness in the writing and the contextualization of findings. Furthermore, as a minor point and as the authors acknowledge, data was collected in 2021 and 3 years later the findings, and there was a risk of non-trivial sampling bias, leaving these findings with some fragility. In terms of writing, there are plenty of minor writing errors, which are easily correctable (e.g. incorrect use of tense), but there are sufficient number of significant writing deficiencies that leave me without confidence that they can be remedied. In totality, I think sufficient work is required here that the manuscript is not ready for publication. My recommendations are twofold. Firstly, I recommend repeating the survey; doing this would provide more data, and more contemporary data as well as the opportunity to conduct a temporal analysis. Secondly, I recommend employing strategies to really tighten up the writing.

Below is some commentary that highlights my concerns but might also assist in any future repurposing of this work.

Sorry to provide a negative review; it comes in good faith and I wish the authors well with this important line of enquiry.

---

- Lines 15-17: Not sure of the logic here. I agree that physiotherapists play a critical role in addressing the impacts of population aging, especially in hospitalized patients. The implied connection made here about the context of workforce shortages though is not clear. Are physiotherapists not vital, independent of workforce shortages? If workforce shortages didn't exist would the role of physiotherapists still not be vital? This just needs a little work in terms of logic progression.

- Line 21: Define FDHRS at its first use for the reader not familiar with the French context.

- Lines 23-25: I am not convinced of the value of this statement. The data doesn't allow for firm interpretation (as the authors acknowledge), particularly given how the (health workforce) world has changed since 2020. Suggest removing.

- Line 40-42: Statement does not make sense. Please review/edit/remove. The narrative works (is better) with the statement simply removed.

- Line 47: Reference for the 30-50% wage gap?

- Lines 48-53: The wording is too strong here. Some suggested improvements: "At the institutional level, there is an opportunity need to develop localized strategies to improve fidelity and attractiveness and to tackle these issues effectively. Given the recruitment challenges in the profession, institutions might consider implementing approaches that emphasize project-based solutions and create a supportive environment for students."

- Lines 39-66: This part of the narrative is confusing, I think due to lack of coherent structure. Paragraphs move backwards and forwards between describing the problem, then suggesting solutions, then returning to the problem again. This makes it hard for the reader to understand. Can I suggest you have a paragraph (or 2) focusing on the mechanisms that explain the recruitment and retention issues. Then, describe the historical strategies that have been employed (and their impact). This should connect the reader more seamlessly to the paragraph starting Line 67.

- Lines 71-72: "How can we implement measures..." This does not make sense. Do you mean strategies? You measure outcomes; you implements strategies (or initiatives, etc.). Please edit wording to better match intended meaning.

- Line 73- "The aim of this study was...

- Lines 75-76: You cite COVID as a contextual factor that may impact attractiveness of employed physiotherapy without referring to it in the background. If it is relevant it should have been introduced prior to here.

- Line 78: "... we aimed..."

- Line 79: "... that may encourage..."

- Lines 80-82: The description of the outcome measure is not necessary or helpful here. Relocate to methods. Edit lines 82-84 for the same reasons as some content here relates more to methods.

- Line 82- "The study examined..."

- Line 85: "we sought to..."

- Line 89: Materials and Methods

- Line 117: "They had..."

- Materials and Methods: I would like to see the entire questionnaire as part of the main manuscript. Reference is made to certain sections in the 'Results', but I think it important to include for reference. Please include and as part of the manuscript rather than as an appendix.

- Table III and Figure 3 present the same data in slightly different ways. I see no reason to have both, the Table is easier to interpret, and the Table is referred to in the main text more than the Figure. Remove either, and I suggest removing Figure 3.

- Line 213: Poor title name

- Lines 224-228 (or even 233): This is a repetition that I don't think is necessary. Suggest removing of trimming significantly.

- Lines 234-241: This paragraph doesn't make sense. You don't frame the paragraph with an introductory sentence, nor any concluding statement. You make various disparate connections between findings and phenomena. Many of which don't make sense. For example, you make a connection between the 'issue of attractiveness' and findings related to low percentage 'feeling overwhelmed'. I simply don't understand this connection. You go on to cite this as a risk factor for of 'psychosocial issues' and then quality of work-life. I appreciate that theoretically there are relationships between many of these phenomena, but you do not make this clear and you do not align theory with your data in a way that might enable the reader to conclude how your data has influenced theory. This paragraph needs substantial re-working.

- Lines 249-252: "... not yet seized the opportunity..." this is political speculation/judgement. I don't disagree with your sentiment but you shouldn't deploy statements such as this or in this way; author bias rings through in its current form. You go on to speculate regarding the impact of 'this failure'. This may be true (or not) but is currently speculative and overstated in the context of your research question and data. This speculation needs tempering.

- Lines 264-269: Once again this paragraph touches briefly on some big issues but without thoughtful explanations and connection to the research question or data. For example, to state 'evidence-based practice' as a change in the profession is simply too broad and vague to be useful. This paragraph needs work.

Reviewer #2: Thank you for the opportunity to review this manuscript ‘Attractiveness and Retention Factors for Employed Physiotherapists in France: A Multicentre Survey’. This observational cross-sectional survey study provides insights into physiotherapists’ perspectives on the attractiveness of working in public health. The authors are to be commended for exploring this important area and achieving a great response to their survey, however some changes are needed to enhance the clarity of this work and its findings.

Overarching comments:

- I have interpreted the term ‘employed physiotherapy’ to mean working as a physiotherapist within the public health sector in France. This terminology requires defining or rephrasing for the manuscript, as the term alone may not convey the same meaning internationally.

- In some sections, the terms ‘attractiveness’ and ‘retention’ are used inconsistently and could be refined. These terms are not always interchangeable, leading to potential confusion. For instance, the abstract primarily mentions ‘attractiveness,’ as does most of the manuscript (e.g., Lines 130-132, which state the domains aimed at ‘improving the attractiveness of employed physiotherapy’). However, the title references ‘retention’ as well. Additionally, Line 99 mentions that the inclusion criteria were designed to ‘gain insights into retention influencing factors.’ Maintaining a consistent focus on ‘attractiveness’ throughout would enhance readability.

Introduction:

- The authors might consider restructuring the introduction. Could the second paragraph serve as a stronger opening for the manuscript?

- Lines 39-42: The statement that government strategies have been in place since 2015 requires more detail. This is mentioned again in the discussion, and providing more context would help readers understand whether your final recommendations align with or differ from previous efforts.

- Are there any geographical challenges to public health employment in France? This aspect is not specifically addressed and could be important to mention.

- The introduction should focus on setting up the background and research question. The explanation of the primary and secondary outcome measures (lines 80-84) is not needed, as it appears (appropriately) in the methods section. (In addition, it does not need repeating in the discussion).

Methods:

- Lines 95-97: The statement regarding the minimum number of responses needs further clarification. This is exploratory observational research conducted via a cross-sectional survey, and no hypotheses are being statistically tested. This is a large sample for survey results, and it can be explored in other ways (e.g. via its characteristics) to argue it is a representative sample of the relevant population.

- I would appreciate more details on the survey development process, particularly how and why the four domains were chosen. Including a copy of the entire survey, with statements related to each domain, would be helpful.

- Lines 117-118: Please clarify how data could be removed upon a participant’s request if it was anonymous. Was the data anonymised after collection, or were all responses anonymous from the start, as suggested in Lines 109-110?

- Lines 141-142: Please revise the term ‘qualitative variable’ as the statements collect Likert scale (categorical) data, not qualitative responses.

- The rationale for analysing age and professional groups is needed. Additionally, how were differences between groups (e.g., by age) determined, and what analysis was used?

- Lines 146-149: Please clarify how the secondary outcomes were examined. The results mention Chi-square analysis, which is suitable for categorical data. What categories were used for ‘age’? Adding a rationale for this analysis would help clarify its connection to your research question.

Results:

- Lines 155-176: The inclusion criteria specify ‘physiotherapists working in public health,’ yet, some managers identified as nurses? Is this correct?

- Please ensure that the units of measurement are clearly indicated in all tables, either in the title or the column headings.

- Does Figure 3 duplicate the information in Table III? Are both necessary? Additionally, the use and interpretation of confidence intervals for Likert scale data need clarification.

- Lines 196-200: Besides clarifying the age categories, this section could elaborate on the findings. For instance, which age categories showed significant relationships? Was an older age associated with higher Likert scale ratings on these items?

- The titles of items 24 and 25 differ between Table IV and lines 196-199. Please ensure consistency.

Discussion:

- This section would benefit from revision. The first two paragraphs summarise the introduction and methods, but the authors could just start with directly summarising their main findings - then expanding on pertinent points.

- Lines 234-241: The authors could clarify how the low proportion of respondents who never felt overwhelmed relates to the issue of attractiveness. Additionally, it is unclear from the results where teamwork, interdisciplinary work, and a sense of belonging were significant. A stronger link to the background and results is needed.

- More detail is needed on how the measures in Table V were derived, including how the study results informed them. For example, the results (Lines 192-195) indicate that mentoring involvement was less significant than work-life balance and professional quality of life, yet measures 7 and 8 focus on supervision and training related to mentoring. More information is needed on what has been previously implemented (e.g. since 2015) and how the suggested measures compare.

- The upgrade of entry-level education to a Master’s degree is mentioned in the introduction (Lines 57-58) and the discussion (Lines 246-247). Not needed as repetition in the discussion.

- A limitation of the study is the use of an unvalidated survey. Also, were participants asked to suggest additional measures or areas influencing attractiveness? (i.e. that weren’t already in the survey questions?) this is another limitation.

- A stronger conclusion that references the research questions and main findings may be needed.

6. PLOS authors have the option to publish the peer review history of their article (what does this mean? ). If published, this will include your full peer review and any attached files.

**Do you want your identity to be public for this peer review?** For information about this choice, including consent withdrawal, please see our Privacy Policy .

Reviewer #1: No

Reviewer #2: No

---

## [Author Response · Author response to Decision Letter 1]

24 Dec 2024

Point by Point response

We thank the reviewers for their valuable comments and suggestions, which have significantly improved the manuscript. Below is a point-by-point response to the reviewers' comments.

The manuscript has been extensively revised to address the reviewers' feedback. Key improvements include enhanced clarity in language, restructuring of sections for better narrative flow, and the addition of details to contextualize findings. Specific changes are outlined below for each reviewer comment.

1. Reviewer #1

1.1. General Comments:

• "Thanks you for the opportunity to review this manuscript. The subject is topical and this study has the potential to be of interest to those interest in health workforce development. Whilst the methods were appropriate, and much of the conduct and reporting of the study are commendable, there is weakness in the writing and the contextualization of findings.."

Response: We have revised the manuscript to address language inconsistencies and improve the narrative structure. Contextualization of findings has been enhanced in the Introduction, Interpretation, and Barriers to Attractiveness and Retention sections of the final manuscript.

• ". Furthermore, as a minor point and as the authors acknowledge, data was collected in 2021 and 3 years later the findings, and there was a risk of non-trivial sampling bias, leaving these findings with some fragility.."

Response: While we acknowledge the importance of updated data, the evaluation of attractiveness measures often requires a decade to fully assess their impact. As such, the conclusions drawn from the 2021 dataset remain relevant and reflective of current challenges. Future studies are planned to include more recent data and enable temporal analysis.

• In terms of writing, there are plenty of minor writing errors, which are easily correctable (e.g. incorrect use of tense), but there are sufficient number of significant writing deficiencies that leave me without confidence that they can be remedied. In totality, I think sufficient work is required here that the manuscript is not ready for publication. My recommendations are twofold. Firstly, I recommend repeating the survey; doing this would provide more data, and more contemporary data as well as the opportunity to conduct a temporal analysis. Secondly, I recommend employing strategies to really tighten up the writing.

Response: We would like to thank reviewer 1 for this comment. We hope that this new version will provide the necessary insights for reviewers and readers of the journal to benefit from our work.

1.2. Specific Comments:

• Lines 15-17: " Not sure of the logic here. I agree that physiotherapists play a critical role in addressing the impacts of population aging, especially in hospitalized patients. The implied connection made here about the context of workforce shortages though is not clear. Are physiotherapists not vital, independent of workforce shortages? If workforce shortages didn't exist would the role of physiotherapists still not be vital? This just needs a little work in terms of logic progression.."

Response: Thank you for your insightful comment. In accordance with your recommendation, we have rewritten the paragraph to clarify the logical progression leading to the issue of workforce shortages. The revised version ensures that the vital role of physiotherapists in addressing the impacts of population aging is highlighted independently of the workforce shortage context, while also logically connecting this role to the broader challenges posed by such shortages. We hope this revision better aligns with your expectations.

• Line 21: " Define FDHRS at its first use for the reader not familiar with the French context.."

Response: Thank you. In accordance with the other comments, the sentence has been rewritten. The source "FDHRS" is now only referenced in the bibliography. Please see line 15 to 46 the corrected Introduction : “This is driven by an aging population requiring more complex care, rising healthcare costs, and an insufficient number of qualified physiotherapists in public sector positions (7–9). These factors exacerbate the strain on public healthcare institutions and underscore the urgent need to address workforce challenges.

The attractiveness and retention of physiotherapists in the public sector have become critical issues in France (2,5). Several factors contribute to these challenges. One major issue is the wage disparity between public and private practice physiotherapists. In 2022, physiotherapists in private practice earned an average of €44,243, 30–50% (10) more than their counterparts employed in the public sector, who earned €32,808 on average (11). This wage gap creates a strong incentive for physiotherapists to choose private practice over public sector employment (12). In addition to financial considerations, public sector physiotherapists often face professional dynamics, management practices, and limited career advancement opportunities, all of which can further deter potential recruits and contribute to high turnover rates (5,13–15).

Moreover, the aging workforce, combined with the increasing number of retirements among experienced physiotherapists, exacerbates the shortage in the public sector (16,17). These factors, alongside the demanding nature of the work and the high patient load in public hospitals, make attractiveness even more challenging.

Between 2016 and 2021, ministerial measures were introduced to enhance salary scales and implement commitment bonuses for public sector employees. However, the effectiveness of these initiatives in addressing attractiveness and retention issues has not been formally assessed (18,19). Despite these efforts, attractiveness and retention challenges persist. The limited success of these measures suggests that a more comprehensive and sustained approach is needed (20,21). Policies have often focused more on nursing staff, with less emphasis placed on physiotherapists, which may have contributed to the ongoing difficulties in the physiotherapy workforce (19). Since 2015, French physiotherapists have been trained over five years, earning a Master’s degree. This reform has led to a substantial increase in their competencies and responsibilities. However, these advancements in qualifications and professional scope have not been matched by corresponding changes in their roles, responsibilities, or compensation within the public sector (22–24). »

• Lines 23-25: "I am not convinced of the value of this statement. The data doesn't allow for firm interpretation (as the authors acknowledge), particularly given how the (health workforce) world has changed since 2020. Suggest removing."

Response: as request, this section has been removed for clarity. Please see line 15 to 46 the corrected Introduction.

• Line 40-42: “Statement does not make sense. Please review/edit/remove. The narrative works (is better) with the statement simply removed.”

Response: thank you. We have removed it.

• Line 47: “Reference for the 30-50% wage gap?”

Response: The references have been added (lines 22-26).

Please see the new references below:

- Les salaires dans la fonction publique hospitalière en 2022 - Insee Première - 2015 [Internet]. [cited 2024 Dec 22]. Available from: https://www.insee.fr/fr/statistiques/8253303

- UNASA. Donnees statistiques sur la profession liberale en France [Internet]. 2021 [cited 2024 Dec 22]. Available from: https://www.unasa.fr/wp-content/uploads/2022/07/2021-Recueil-UNASA.pdf

- Flodgren G, Eccles MP, Shepperd S, Scott A, Parmelli E, Beyer FR. An overview of reviews evaluating the effectiveness of financial incentives in changing healthcare professional behaviours and patient outcomes - Flodgren, G - 2011 | Cochrane Library. [cited 2024 Dec 22]; Available from: https://www.cochranelibrary.com/cdsr/doi/10.1002/14651858.CD009255/abstract

• Lines 48-53: The wording is too strong here. Some suggested improvements: "At the institutional level, there is an opportunity need to develop localized strategies to improve fidelity and attractiveness and to tackle these issues effectively. Given the recruitment challenges in the profession, institutions might consider implementing approaches that emphasize project-based solutions and create a supportive environment for students."

Response: Thank you for this suggestion, which we have added to the heavily modified introduction text.

• Lines 39-66: " This part of the narrative is confusing, I think due to lack of coherent structure. Paragraphs move backwards and forwards between describing the problem, then suggesting solutions, then returning to the problem again. This makes it hard for the reader to understand. Can I suggest you have a paragraph (or 2) focusing on the mechanisms that explain the recruitment and retention issues. Then, describe the historical strategies that have been employed (and their impact). This should connect the reader more seamlessly to the paragraph starting Line 67.."

Response: In response to your comment, we have restructured the section to enhance coherence and improve the narrative flow. Following your suggestion, we have dedicated one paragraph to discussing the mechanisms underlying recruitment and retention issues, followed by a separate paragraph outlining the historical strategies employed and their impacts. These changes aim to create a more seamless transition to the paragraph. Please see line 15 to 46 the corrected Introduction.

• Lines 71-72: ""How can we implement measures..." This does not make sense. Do you mean strategies? You measure outcomes; you implements strategies (or initiatives, etc.). Please edit wording to better match intended meaning..'"

Response: The wording has been revised for clarity. The terms "measures" and "strategies" have been used in the manuscript with specific intent: "measures" refers to specific actions or policies, while "strategies" denotes broader, overarching approaches. This distinction has been maintained throughout for clarity.

• Lines 75-76: You cite COVID as a contextual factor that may impact attractiveness of employed physiotherapy without referring to it in the background. If it is relevant it should have been introduced prior to here.

Response: thank you for your feedback. covid does not seem relevant. we have removed this referring.

• Line 78: " The aim of this study was....'"

Response: Corrected for grammatical accuracy. (See Introduction, line 47.)

• COVID Context (Lines 75-76):"Introduce COVID earlier or remove."

Response: References to COVID-19 have been removed to maintain focus on the primary objective.

• Line 79: "... that may encourage..."

Response: it is corrected.

• Lines 80-82: The description of the outcome measure is not necessary or helpful here. Relocate to methods. Edit lines 82-84 for the same reasons as some content here relates more to methods.

Response: it’s done.

• Line 82- "The study examined..."

Response: it’s corrected.

• Line 85: "we sought to..."

Response: it’s corrected.

• Line 89: Materials and Methods

Response: it’s corrected.

• Line 117: "They had..."

Response: it’s corrected.

• Materials and Methods: " I would like to see the entire questionnaire as part of the main manuscript. Reference is made to certain sections in the 'Results', but I think it important to include for reference. Please include and as part of the manuscript rather than as an appendix.

Response: A table summarizing survey questions has been added (please see line 107 to 108 in Materials and Methods session, Table I). The table is presented below.

Table I: Questionnaire of the survey

PART I: General demographic questions

PART II: Feelings and Experiences in Daily Work

Question: "Do you agree with the following statement?"

Response options: "Strongly agree," "Agree," "Neutral," "Disagree," "Strongly disagree”.

1. My institution faces an issue with the attractiveness of physiotherapist positions.

2. The attractiveness measures implemented by successive governments since 2016 have been sufficient.

3. You experience difficulties or feel overwhelmed in your daily work.

PART III: Opinion About Attractiveness Measures

Question: "Do you agree with the following statement?"

Response options: "Strongly agree," "Agree," "Neutral," "Disagree," "Strongly disagree."

Domain 1: Adoption of National Measures

1. Wage increases enhance employee attractiveness.

2. Recognizing a Master's degree for physiotherapy studies enhances employee attractiveness.

3. Recognizing additional diplomas in the public hospital service enhances employee attractiveness.

Domain 2: Inclusion in Daily Structure

4. Including physiotherapists in the development of rehabilitation projects enhances employee attractiveness.

5. Including physiotherapists in shared inter-facility projects enhances employee attractiveness.

6. Including physiotherapists in the management of their facility enhances employee attractiveness.

Domain 3: Supervision of Employed Physiotherapists

7. Having a healthcare manager from one of the rehabilitation professions enhances employee attractiveness.

8. Having a senior healthcare executive from one of the rehabilitation professions enhances employee attractiveness.

9. Entrusting cross-functional activities to physiotherapists (e.g., ethics, steering commissions, working groups) enhances employee attractiveness.

Domain 4: Quality of Life and Mentorship

10. Involving physiotherapists in tutoring trainees enhances employee attractiveness.

11. The quality of work life is an important factor in choosing your professional activity.

12. Management consideration of personal quality of life promotes employee attractiveness.

• Figures and Tables: " Table III and Figure 3 present the same data in slightly different ways. I see no reason to have both, the Table is easier to interpret, and the Table is referred to in the main text more than the Figure. Remove either, and I suggest removing Figure 3.."

Response: Figure 3 has been removed. Please see Results, line 165.

• Line 213: Poor title name

Response: we have corrected it by the following title: “Effect of age and current profession on socio-professional outcomes (n=448)”

• Lines 224-228 (or even 233): This is a repetition that I don't think is necessary. Suggest removing of trimming significantly.

Response: it’s deleted.

• Lines 234-241: " This paragraph doesn't make sense. You don't frame the paragraph with an introductory sentence, nor any concluding statement. You make various disparate connections between findings and phenomena. Many of which don't make sense. For example, you make a connection between the 'issue of attractiveness' and findings related to low percentage 'feeling overwhelmed'. I simply don't understand this connection. You go on to cite this as a risk factor for of 'psychosocial issues' and then quality of work-life. I appreciate that theoretically there are relationships between many of these phenomena, but you do not make this clear and you do not align theory with your data in a way that might enable the reader to conclude how your data has influenced theory. This paragraph needs substantial re-working.."

Response: The paragraph has been rewritten for clarity and coherence. It now begins with an introductory sentence explaining the connection between the "issue of attractiveness" and psychosocial risks. The relationship between feeling overwhelmed and quality of work-life has been better aligned with the data and contextualized within relevant theories. A concluding sentence ties these findings to the broader discussion on workforce challenges. Please see the corrected Discussion line 234 to 240:

“The study revealed several barriers that reduce the attractiveness of employed physiotherapy positions. Inadequate compensation and lack of recognition for advanced qualifications emerged as critical issues. Respondents under 40 years old were particularly sensitive to the lack of career development opportunities and the limited integration of physiotherapists into institutional decision-making. These findings align with prior literature emphasizing the importance of aligning professional roles with advanced training and qualifications (35–37).”

• Lines 249-252: "."...not yet seized the opportun

---

## [Decision Letter · Decision Letter 1]

18 Feb 2025

PONE-D-24-30478R1Attractiveness and Retention Factors for Employed Physiotherapists in France: A Multicentre SurveyPLOS ONE

Dear Dr. Sarhan,

Thank you for submitting your manuscript to PLOS ONE. After careful consideration, we feel that it has merit but does not fully meet PLOS ONE’s publication criteria as it currently stands. Therefore, we invite you to submit a revised version of the manuscript that addresses the points raised during the review process.

We look forward to receiving your revised manuscript.

Kind regards,

Saravana Kumar

Academic Editor

PLOS ONE

Journal Requirements:

Reviewers' comments:

Reviewer's Responses to Questions

**Comments to the Author**

1. If the authors have adequately addressed your comments raised in a previous round of review and you feel that this manuscript is now acceptable for publication, you may indicate that here to bypass the “Comments to the Author” section, enter your conflict of interest statement in the “Confidential to Editor” section, and submit your "Accept" recommendation.

Reviewer #1: All comments have been addressed

Reviewer #2: (No Response)

2. Is the manuscript technically sound, and do the data support the conclusions?

Reviewer #1: Yes

Reviewer #2: Yes

3. Has the statistical analysis been performed appropriately and rigorously? 

Reviewer #1: N/A

Reviewer #2: I Don't Know

4. Have the authors made all data underlying the findings in their manuscript fully available?

Reviewer #1: Yes

Reviewer #2: Yes

5. Is the manuscript presented in an intelligible fashion and written in standard English?

Reviewer #1: Yes

Reviewer #2: Yes

6. Review Comments to the Author

Reviewer #1: (No Response)

Reviewer #2: I would like to commend the authors on their consideration of the previous comments and their efforts in reviewing the manuscript. It is much improved - particularly the introduction.

I have just a couple of very minor points to raise:

The term 'employed physiotherapy' still persists in some text - e.g. in discussion and title Table IV

Table III presents results of Chi-square analysis - yet, the categories used for this analysis are not clear. The text mentions 'participants <40 years' - were <40 and >40 years categories used for age in this analysis? Presenting the p-value next to the variable, doesn't communicate the analysis and which category was significantly different.

Line 126 'This four...' needs correcting

Discussion - I suggest moving the 'limitations' to before the conclusions rather than before the interpretation of findings.

Lines 272-276 - This section states that financial revaluations and attractiveness bonuses have had limited success. Yet, revising salaries is the first measure the authors suggest. Are the authors suggesting profession-wide policy change is needed here to align all public institutions - if so, this needs to be clearer.

7. PLOS authors have the option to publish the peer review history of their article (what does this mean? ). If published, this will include your full peer review and any attached files.

**Do you want your identity to be public for this peer review?** For information about this choice, including consent withdrawal, please see our Privacy Policy .

Reviewer #1: No

Reviewer #2: No

---

## [Author Response · Author response to Decision Letter 2]

24 Feb 2025

Point-by-Point Response to Reviewers

Manuscript ID: PONE-D-24-30478R1

Title: Attractiveness and Retention Factors for Employed Physiotherapists in France: A Multicentre Survey

Dear Editor and Reviewers,

We would like to express our gratitude to the reviewers and the editor for their comments, which have helped us improve the manuscript. Below, we provide detailed responses to each point raised. All modifications have been incorporated into the revised manuscript, with track changes.

Reviewer #1

Comment: All comments have been addressed.

Response: We appreciate the reviewer's positive feedback and are pleased that the revisions have addressed their concerns.

Reviewer #2

General Comment

"I would like to commend the authors on their consideration of the previous comments and their efforts in reviewing the manuscript. It is much improved - particularly the introduction."

Response: Thank you for your positive feedback. We appreciate your comments and have carefully addressed the remaining minor issues to further improve the clarity and coherence of the manuscript.

Comment 1: Persistence of "employed physiotherapy" in some sections

"The term 'employed physiotherapy' still persists in some text - e.g., in discussion and title Table IV."

Response: We have carefully reviewed the manuscript and replaced all instances of "employed physiotherapy" with a more precise and appropriate term. The title of Table IV has also been updated accordingly.

Line 215: “The results provide a nuanced understanding of the factors influencing the attractiveness of physiotherapy positions in public hospitals.”

Line 231: “The study revealed several barriers that diminish the appeal of physiotherapy positions in public hospitals.”

Line 266: “Table VI: 10 measures to improve the appeal of physiotherapy positions in public hospitals.”

Comment 2: Clarification of Chi-square analysis in Table III

"Table III presents results of Chi-square analysis - yet, the categories used for this analysis are not clear. The text mentions 'participants <40 years' - were <40 and >40 years categories used for age in this analysis? Presenting the p-value next to the variable doesn’t communicate the analysis and which category was significantly different."

Response: We have clarified the age categories used in the Chi-square analysis by explicitly stating that the groups were "<40 years" and ">=40 years". Additionally, we have added the corresponding Chi-square values (χ²) in Table III to provide a clearer representation of the statistical analysis. To improve readability, we have also removed the "current profession" (manager) category from the table, as the results were not significant.

Line 193: “Conversely, no significant associations were observed between others measured variables”

Line 196:

Table V : Effect of age on socio-professional outcomes (n=448). For statistical analysis, age was dichotomized into two categories: <40 years and ≥40 years. The chi-square test was used to assess differences between these groups regarding the stated variables.

The quality of worklife is important in the choice of your professional activity

Age group

n (%) Strongly agree Agree Neither disagree nor agree Disagree Strongly disagree Chi-Squared Tests

<40 221 (89.5%) 25 (10.1%) 1 (0.4%) 0 (0.0%) 0 (0.0%) Χ² = 27.4

p < .001

≥40 141 (70.1%) 56 (27.9%) 2 (1.0%) 2 (1.0%) 0 (0.0%)

Taking personal quality of life into account by management promotes attractiveness

<40 177 (71.7%) 62 (25.1%) 8 (3.2%) 0 (0.0%) 0 (0.0%) Χ² = 17.2

p = 1.7×10-3

≥40 109 (54.2%) 76 (37.8%) 13 (6.5%) 2 (1.0%) 1 (0.5%)

Comment 3: Correction of Line 126

"Line 126 'This four...' needs correcting."

Response: This typographical error has been corrected in the revised manuscript.

Line 104: “These four domains were chosen”

Comment 4: Placement of the "Limitations" section

"Discussion - I suggest moving the 'limitations' section to before the conclusions rather than before the interpretation of findings."

Response: We have followed the reviewer's suggestion and moved the "Limitations" section to appear before the conclusions: line 168.

Comment 5: Clarification on financial revaluations and salary revisions (Lines 272-276)

"Lines 272-276 - This section states that financial revaluations and attractiveness bonuses have had limited success. Yet, revising salaries is the first measure the authors suggest. Are the authors suggesting profession-wide policy change is needed here to align all public institutions - if so, this needs to be clearer."

Response: We acknowledge the need for clarification and have revised this section to explicitly state that salary adjustments should be considered as part of a broader structural reform aimed at ensuring consistency across public institutions. The revised text now highlights that while previous financial incentives had limited success, a systematic and profession-wide policy change could enhance long-term attractiveness by addressing disparities in remuneration and career progression.

Line 243: “While financial revaluations and attractiveness bonuses have had limited success when implemented in isolation, they remain essential as part of a broader, systemic reform. A structured salary revision, combined with institutional policies recognizing additional qualifications and improving working conditions, could help address the deeper structural challenges of the profession.”

Additional Journal Requirements

We have also reviewed and updated the reference list to ensure completeness and accuracy. No retracted papers have been cited.

We hope that these revisions adequately address all concerns. Please let us know if further modifications are needed.

Kind regards,

---

## [Editor Report · Decision Letter 2]

27 Feb 2025

Attractiveness and Retention Factors for Employed Physiotherapists in France: A Multicentre Survey

PONE-D-24-30478R2

Dear Dr. Sarhan,

We’re pleased to inform you that your manuscript has been judged scientifically suitable for publication and will be formally accepted for publication once it meets all outstanding technical requirements.

Kind regards,

Saravana Kumar

Academic Editor

PLOS ONE
---

## [Editor Report · Acceptance letter]

PONE-D-24-30478R2

PLOS ONE

Dear Dr. Sarhan,

I'm pleased to inform you that your manuscript has been deemed suitable for publication in PLOS ONE. Congratulations! Your manuscript is now being handed over to our production team.

Kind regards,

on behalf of

Professor Saravana Kumar

Academic Editor

PLOS ONE